# Sudden Unexpected Death Associated with Arrhythmogenic Cardiomyopathy: Study of the Cardiac Conduction System

**DOI:** 10.3390/diagnostics11081323

**Published:** 2021-07-23

**Authors:** Giulia Ottaviani, Graziella Alfonsi, Simone G. Ramos, L. Maximilian Buja

**Affiliations:** 1Lino Rossi Research Center, Anatomic Pathology, Department of Biomedical, Surgical and Dental Sciences, Università Degli Studi di Milano, 20122 Milan, Italy; Graziella.Alfonsi@unimi.it; 2Department of Pathology and Laboratory Medicine, McGovern Medical School, University of Texas Health Science Center at Houston (UTHealth), Houston, TX 77030, USA; L.Maximilian.Buja@uth.tmc.edu; 3Pathology and Forensic Medicine, Ribeirão Preto Medical School, University of São Paulo, Ribeirão Preto 14049-900, Brazil; sgramos@fmrp.usp.br

**Keywords:** arrhythmogenic cardiomyopathy, arrhythmogenic right ventricular cardiomyopathy, cardiac conduction system, sudden unexpected cardiac death, autopsy

## Abstract

A retrospective study was conducted on pathologically diagnosed arrhythmogenic cardiomyopathy (ACM) from consecutive cases over the past 34 years (*n* = 1109). The anatomo-pathological analyses were performed on 23 hearts diagnosed as ACM (2.07%) from a series of 1109 suspected cases, while histopathological data of cardiac conduction system (CCS) were available for 15 out of 23 cases. The CCS was removed in two blocks, containing the following structures: Sino-atrial node (SAN), atrio-ventricular junction (AVJ) including the atrio-ventricular node (AVN), the His bundle (HB), the bifurcation (BIF), the left bundle branch (LBB) and the right bundle branch (RBB). The ACM cases consisted of 20 (86.96%) sudden unexpected cardiac death (SUCD) and 3 (13.04%) native explanted hearts; 16 (69.56%) were males and 7 (30.44%) were females, ranging in age from 5 to 65 (mean age ± SD, 36.13 ± 16.06) years. The following anomalies of the CCS, displayed as percentages of the 15 ACM SUCD cases in which the CCS has been fully analyzed, have been detected: Hypoplasia of SAN (80%) and/or AVJ (86.67%) due to fatty-fibrous involvement, AVJ dispersion and/or septation (46.67%), central fibrous body (CFB) hypoplasia (33.33%), fibromuscular dysplasia of SAN (20%) and/or AVN (26.67%) arteries, hemorrhage and infarct-like lesions of CCS (13.33%), islands of conduction tissue in CFB (13.33%), Mahaim fibers (13.33%), LBB block by fibrosis (13.33%), AVN tongue (13.33%), HB duplicity (6.67%%), CFB cartilaginous meta-hyperplasia (6.67%), and right sided HB (6.67%). Arrhythmias are the hallmark of ACM, not only from the fatty-fibrous disruption of the ventricular myocardium that accounts for reentrant ventricular tachycardia, but also from the fatty-fibrous involvement of CCS itself. Future research should focus on application of these knowledge on CCS anomalies to be added to diagnostic criteria or at least to be useful to detect the patients with higher sudden death risks.

## 1. Introduction

Arrhythmogenic cardiomyopathy (ACM), formerly known as arrhythmogenic right ventricular cardiomyopathy (ARVC), is an inheritable type of cardiomyopathy not secondary to ischemic, hypertensive, or valvular heart disease characterized by ventricular fibro-fatty myocardial replacement and prominent scar-related arrhythmias. Clinically, ACM presents increased risk for palpitations, syncope, malignant ventricular arrhythmias, or sudden unexpected cardiac death (SUCD) without any recognized preceding cardiac dysfunction, especially in young athletes. Pathologically, ACM is characterized by progressive loss of myocardium replaced by fibro-fatty tissue, especially in the right ventricle. ACM, as an evolution of the original term ARVC, is a biventricular muscle disease with a similar or even exceeding involvement of the left ventricle compared to the involvement of the right ventricle [1,2,3,4,5,6].

Genetically, ACM is a hereditary Mendelian desmosome disease, typically dominant and rarely recessive, with mutations of genes encoding intercellular proteins. Twelve ARVC loci, ARVC1–ARVC12, have been described [7,8]. Most of the pathogenic variants in ACM have been identified in genes encoding the intercalated disk (ID) proteins junction plakoglobin (JUP), desmoplakin (DSP), plakophilin-2 (PKP2), desmoglein-2 (DSG2), desmocollin-2 (DSC2), α-T-catenin (CTNNA3), and N-cadherin (CDH2). A few other mutations have been found in non-ID genes encoding cardiac ryanodine receptor 2 (RYR2), transforming growth factor β3 (TGFβ3), transmembrane protein 43 (TMEM43), desmin (DES), titin (TTN), lamin A/C (LMNA), phospholamban (PLN), sodium voltage-gated channel α subunit 5 (SCN5A), and tight junction protein 1 (TJP1) [9]. Recently, an international multidisciplinary ARVC Clinical Genome Resource Gene Curation Expert Panel [10] reappraised all reported ARVC genes. The genes that have strong, moderate, or low association with ACM or ARVC have been highlighted. Of 26 reported ARVC genes, six genes, *PKP2*, *DSP*, *DSG2*, *DSC2*, *JUP*, and *TMEM43*, have strong evidence for ARVC causation. Two genes, *DES* and *PLN*, have moderate evidence for ARVC. The remaining 18 genes, such as *RYR2*, had limited or no evidence for ARVC.

The true prevalence of ACM is unknown and has been estimated to be at the rate of 1:1000 of the general population. Considering that SUCD can be the initial manifestation of ACM, undiagnosed ACM patients probably would increase its actual prevalence of an additional 30% [1].

In 1982, the first report by Marcus et al. [11] described 24 patients affected by a dysplasia of the right ventricular myocardium thought to be a congenital developmental defect. The term “triangle of dysplasia” was introduced the most frequent sites of dysplasia: (1) anterior infundibulum, (2) right ventricular apex and (3) inferior or diaphragmatic aspect of the right ventricle. In 1996, the definition and classification of cardiomyopathies by the World Health Organization (WHO) [12] included the ARVC as a newly recognized form of cardiomyopathy. In 2006, the American Heart Association (AHA) [13] grouped the cardiomyopathies into genetic, mixed, and acquired forms, classifying the ARVC as primary cardiomyopathy of genetic type. In 2008, the European Society of Cardiology (ESC) [14] divided the ARVC into the familial or genetic and nonfamilial or nongenetic forms. 

Recently, ARVC has been categorized as arrhythmogenic cardiomyopathy (ACM). ACM is a cardiomyopathy characterized by fibro-fatty replacement of the right ventricle, of both ventricles or of the left ventricle which may even exceed the severity of right ventricle involvement. Currently, the classification of ACM remains controversial and it is questioned whether ACM is a distinct cardiomyopathy, previously defined as ARVC [1], rather than a morphologic trait shared by a broad spectrum of cardiomyopathies, including ARVC, left ventricular noncompaction, cardiac amyloidosis, sarcoidosis, and Chagas disease [3]. The characterization of ACM has been made by autopsy investigations, genotype-phenotype correlations and the increasing use of contrast-enhancement cardiac magnetic resonance (CE-CMR) [2]. Autopsy investigation has its limitations as well. For example, there are still no definite criteria to diagnose ACM and there is still some confusion about the use of the term ACM rather than ARVC and if the diagnostic criteria at autopsy refer to which one of the two or to both. Moreover, other diseases are in differential diagnosis with ACM. To confirm the correct diagnosis, it is useful to evaluate the presence of apoptosis, to quantify the residual myocytes that should be <60%, and to evaluate the hypertrophy of cardiomyocytes [15,16].

The latest Heart Disease and Stroke Statistics—2021 Updated by the AHA [8] describes only ARVC, without reporting any specific epidemiological data on ACM. A recent report by the same group that introduced the ACM term as the natural evolution of ARVC [6], still uses the term of ARVC [17], assessing the prevalence of left ventricular (LV) involvement and features of LV phenotype (ARVC-LV).

Although it is known that ACM is related to lethal arrhythmias and that ACM patients often die suddenly and unexpectedly, little, if anything, is known about the possible histopathological abnormalities and/or the involvement of the cardiac conduction system in ACM patients. 

The aim of this study is to determine the presence and significance of the possible abnormalities or involvement of the cardiac conduction system in ACM. 

## 2. Materials and Methods 

### 2.1. Study Population and Data Collection

A total of 1109 cases submitted to the Lino Rossi Research Center of the Università degli Studi di Milano, Milan, Italy, for in-depth specialist post-mortem investigations. These cases were collected from January 1987 to March 2021 and were retrospectively reviewed. Age, gender, ethnicity, history, and circumstances of death, or heart transplant, were recorded, as provided by the referring pathologists or coroners. The privacy and confidentiality in personal data collection and processing were ensured, in accordance with European and U.S. legislation. The institutional review board (IRB) of the Università degli Studi di Milano reviewed and approved this research project on 20 November 2020, providing funding support PSR 2020, linea 2.

After exclusion of violent causes of death, a case was classified as Sudden Unexpected Death (SUD) when the death occurred suddenly within 24h of the beginning of symptoms and unexpectedly by history. 

A case of SUD was classified post-mortem as Sudden Unexpected Cardiac Death (SUCD) when the death could not be related to a noncardiac cause of death.

A total of 38 consecutive cases referred to the Lino Rossi Center were retrospectively enrolled into the study, if they met the pathological criteria to be divided in the two following groups, for comparison: (1) ACM group: 23 cases; (2) Non-ACM group: 15 cases.

### 2.2. Necropsy Investigational Protocol

Hearts were examined macroscopically and microscopically, according to the protocols devised by the Lino Rossi Research Center, Anatomic Pathology, Università degli Studi di Milano, Milan, Italy, including in particular, an in-depth histopathological examination of the cardiac conduction system on serial sections [18,19,20]. 

For the macroscopic study of the hearts, the heart weight, diameters, and wall thickness of the ventricles and interventricular septum were recorded. The thickness of the chamber walls was analyzed by transillumination, exposing the hearts against a fairly intense light source. Each heart was examined for pathological changes in the pericardium, atria, ventricles, septa, cardiac chambers, valves, and coronary arteries. Attention was given to the inspection of the so-called “triangle of dysplasia” [11]. Multiple samples of the major coronary arteries were collected and examined. 

Samples of full-thickness of myocardium were removed for histopathologic examination from the right and left atria and ventricles, from the ventricular septum, mostly parallel to the long axis of the heart. The heart samples were stained with Hematoxylin–Eosin (HE) and Trichromic Heidenhain (Azan).

For the morphological study of the conduction system, two block specimens of the heart were obtained for paraffin embedding. (1) The first block was centered upon the *sulcus-crista terminalis* and contained the sino-atrial node (SAN), its atrial approaches, and the SAN gangliar plexus. At heart sampling, two longitudinal cuts are driven, parallel to the sulcus-crista line, through the atrial wall with a medial prolongation on the right side to encompass the anterior aspect of the inlet of the superior vena cava. On the left side, the cava-cava bridge is sectioned medially, prolonging the cut on the superior vena cava wall. (2) The second block was centered upon the *pars membranacea septi* and contained the atrio-ventricular junction (AVJ) including the atrio-ventricular node (AVN), the His bundle (HB), the bifurcation (BIF), the left bundle branch (LBB) and the right bundle branch (RBB). At heart sampling, the following cuts are driven: an inferior, longitudinal incision through the posterior part of the septum, across the AV annulus fibrous and up to the superior margin of the coronary sinus ostium; an anterior longitudinal incision parallel to the former, through the superior part of the septum, extending to the aortic valvular ring; and two cuts perpendicular to the previous two cuts. The two blocks were routinely fixed in 10% buffered formalin and embedded in paraffin. The sections were cut at intervals of 20–40 μm (levels). For each level, three 8-μm sections are retained, mounted, and, at alternate levels, stained with HE and Azan. All intervening sections are kept and stained as deemed necessary [20,21]. 

Hearts were diagnosed with ACM if they exhibited transmural loss of myocardial cells with fibro-fatty replacement, either regional or diffuse, of the right ventricle and in the absence of significant coronary artery disease or other known causes, with or without involvement of the left ventricle and/or the interventricular septum.

In a study case, according to a referring institution, genetic testing was performed from blood using gene sequencing with deletion and duplication analysis for a panel of the following 68 genes related with Sudden Death Syndrome: *AKAP9*, *ANK2*, *CACNA1C*, *CACNB2*, *CALR3*, *CASQ2*, *CAV3*, *CSRP3*, *CTF1*, *DES*, *DSC2*, *DSG2*, *DSP*, *DTNA*, *EYA4*, *FBN1*, *FBN2*, *FKTN*, *GJA5*, *GPD1L*, *JPH2*, *JUP*, *KCNA5*, *KCNE1*, *KCNE2*, *KCNE3*, *KCNH2*, *KCNJ2*, *KCNQ1*, *LAMP2*, *LDB3*, *LMNA*, *LRP6*, *MYBPC3*, *MYH6*, *MYH7*, *MYL2*, *MYL3*, *MYLK2*, *MYOZ2*, *NEXN*, *NPPA*, *PKP2*, *PLN*, *PRKAG2*, *PSEN1*, *PSEN2*, *RBM20*, *RYR2*, *SCN1B*, *SCN3B*, *SCN4B*, *SCN5A*, *SGCD*, *SLC25A4*, *SNTA1*, *TAZ*, *TCAP*, *TGFB3*, *TGFBR2*, *TMEM43*, *TMPO*, *TNNC1*, *TNNI3*, *TNNT2*, *TPM1*, *TTN*, *VCL* (Fulgent Genetics, Inc., Temple City, CA, USA).

### 2.3. Statistical Analysis

Quantitative data were expressed as means ± SD. The significance of differences between group parameters was evaluated by Student’s *t*-test, χ2 test, or Fisher’s test. In case of skewed distribution, a nonparametric Whitney rank sum test was used. The statistical software SigmaStat^®^ (version 4, Systat Software Inc., Chicago, IL, USA) and SigmaPlot ^®^ (version 14.5, Systat Software Inc., Chicago, IL, USA) were used. The selected level of significance was *p* < 0.05, two-tailed.

## 3. Results

### 3.1. Patients’ Characteristics

Of a series of 1109 cases submitted to the Lino Rossi Research Center of the Università degli Studi di Milano, Milan, Italy, for in depth- specialized anatomo-pathological investigations, collected from January 1987 to 23 April 2021 (2.07%) submitted hearts were diagnosed anatomo-pathologically as arrhythmogenic cardiomyopathy (ACM). Of the total 1109 referred cases, 823 post-mortem cases were classified as sudden unexpected cardiac death (SUCD). In this SUCD group, including an age range from 22 gestational weeks to 90 years, we identified 20 (2.43%) ACM cases. Within the age range of 1 to 90 years, the ACM cases were 20 (5.22%) out of 383 SUCD cases. 

Among the 23 ACM cases, 20 (86.9%) were victims of SUCD and 3 (13.04%) were native explanted hearts from recipients of orthotopic heart transplant (OHT); 16 (69.56%) were males and 7 (30.44%) were females, ranging in age from 5 to 65 (mean age ± SD, 36.13 ± 16.06) years (Figure 1) (Table 1). 

The age difference at ACM diagnosis between males (32.94 ± 14.42) and females (43.43 ± 18.36) was not statistically significant (*p* = 0.154). Most of the cases were diagnosed at the fourth decade of life. Among the two children younger than 10 years of age, one was a 9-year-old boy, and the youngest child was a 5-year-old girl (Figure 1).

Twenty-two (95.65%) patients were Caucasians and one (4.35%) was African American (Table 1). The body mass index (BMI) was referred to be within normal ranges, except in one patient, African American, otherwise considered in good health, whose BMI was 32.4 Kg/m^2^. 

In our 34-year retrospective study of 23 consecutive cases of ACM, death occurred suddenly and unexpectedly in 20 (86.96%) cases. The sudden death occurred with or without witness, as follows: at rest: sitting in seven (30.44%) cases, during sleep in two (8.69%) cases, in a street or another public location in two (8.69%) cases, at the hospital in two (8.69%) cases; at activities: during sports exercise in two (8.69%) cases, during recreational activity in two (8.69%) cases, and while driving in one (4.35%) case. In two (8.69%) cases, the circumstances of death were unknown. The remaining three (13.04%) cases were native explanted hearts, removed at the time of OHT, in patients suffering from congestive heart failure (CHF) (Table 1).

Four (17.4%) patients had a familial occurrence of SUCD below the age of 50 years, but only one (4.35%) patient had a family member known to be affected by ACM. 

According to the available information, only one (4.35%) of our 23 ACM cases was a competitive athlete. In three (13.04%) cases, the SUCD was preceded by overlooked warning symptoms, such as arrhythmia-related symptoms, palpitations and syncopal episodes, that were marginally investigated or were considered to be anxiety-related. One (4.35%) patient had a history of mitral valve prolapse. One (4.35%) patient, 5-year-old girl previously diagnosed with Dandy-Walker syndrome, was found unresponsive during hospital stay after undergoing surgery to place a ventriculoperitoneal shunting catheter. One (4.35%) patient had a history of myocardial infarction for which the patient presented with ventricular fibrillation at emergency room, 10-year prior to death. One (4.35%) patient had an increased body mass index (BMI) of 32.4 Kg/m^2^ in absence of other known pathology. Three (13.04%) patients were diagnosed with heart failure and received OHT. In 13 (56.52%) subjects, the SUCD was the definite first manifestation of ACM as the subjects never experienced any heart-related symptoms nor were known to suffer from any other disease. 

We were informed of only two (8.69%) families within our 23 cases who underwent genetic tests to identify possible genetic anomalies within family members. The children of one case tested negative for genetic anomalies, while the daughter of another case presented a variant of the *DSP* gene. Two family members, a daughter and a brother, of the deceased ones were treated with implantable defibrillators. Overall, the clinical information accompanying our cases were more than scarce, despite the efforts of the first author to gather more data. 

An electrocardiogram (ECG) was available for 6 (26.09%) patients and showed ventricular fibrillation in two patients; low QRS voltages in two patients; epsilon waves in one patient; and an incomplete right bundle block in one patient. In the latest case, a diagnosed of Brugada syndrome was confirmed by a positive Ajmaline test (Table 1).

Post-mortem genetic tests were provided by the referring center for one (4.35%) case and demonstrated a heterozygous pathogenic mutation in gene *PKP2*, a clinically significant variant of autosomal dominant and recessive inheritance, consistent with the autopsy diagnosis of ACM [7]. In the same case, two variants in the genes *MYBPC3* and *AKAP9* of autosomal dominant inheritance, of potential clinical relevance, were identified.

### 3.2. Anatomo-Pathological Findings of the Heart

Of the 23 hearts diagnosed anatomo-pathologically as arrhythmogenic cardiomyopathy (ACM), 20 were from autopsy study and three were native hearts, explanted at time of heart transplantation.

#### 3.2.1. Macroscopic Findings

Macroscopically, all the ACM hearts showed global dilatation of the right ventricle (RV) cavity, with transmural muscle loss and thinning of the RV wall (Figure 2), confirmed by transillumination placing the heart wall against a source of light (Figure 3). The heart weight mean ± SD was 410.33 ± 120.67 g, with a range from 92 (in a 5-year-old child) to 664 g. A concomitant cardiac anomaly was observed only in one (4.35%) case that presented a double origin of the right coronary artery from the proper Valsalva sinus.

The RV chamber was generally dilated, presenting bulging and/or aneurysms, accompanied with a thinned RV wall, characterized by a fatty scarring pattern (Figure 2). The RV thickness had a mean ± SD of 4.57 ± 2.06, ranging from 1 to 8 mm. The RV free wall was generally markedly thinned and translucent at transillumination (Figure 3). In 9 (39.13%) hearts the RV wall thickness was preserved. 

The left ventricle (LV) thickness had a mean ± SD of 17.08 ± 3.25, ranging from 13 to 25 mm. The interventricular septum (IVS) thickness had a mean ± SD of 16.23 ± 3.0, ranging from 9 to 20 mm. 

#### 3.2.2. Microscopic Findings

Histopathological examination disclosed decreased thickness of the wall of the RV, transmural myocardial cell loss with adipose and fibrous tissue replacement in the RV, in absence of significant coronary artery disease or other known cardiac causes, in all cases (100%). The RV was mostly filled with fatty tissue mixed with fibrous tissue except for the surviving myocardium that was normal or sometimes showed degenerative changes, such as intracytoplasmic vacuoles and myofibrillar loss. The adipose tissue was either disorganized or sometimes had a columnar pattern (Figure 4). The fibrous tissue was interstitial fibrosis or replacement fibrosis. Papillary muscles mostly remained normal (78.26% of cases) or were involved by the fatty and fibrous replacement (21.74% of cases).

Five (21.74%) patients had isolated RV involvement, accompanied by unremarkable histology of the interventricular septum (IVS) and left ventricle (LV) (Figure 2D,E). Myocardial replacement by fat and fibrous tissue involved the LV in 18 (78.26%) cases (Figure 5), and the IVS in 13 (56.52%) cases. Thickening of the LV and IVS, irregular arrangement of myocardial fibers (myocyte disarray) and hypertrophy pointed to concomitant hypertrophic cardiomyopathy in 3 (13.04%) cases, although there were other regions in the myocardium without these changes. Foci of anisoinotropism characterized by contraction bands associated with coagulation necrosis, respectively in hyperacute and acute myocardial infarction, were observed in 7 (30.44%) cases (Figure 5B,C). 

Fibro-fatty infiltration, similar to the one detected in the RV, was also detected in the right atrium (RA) in 20 (86.96%) cases (Figure 4E,F) and in the left atrium (LA) in 10 (43.48%) cases.

The adipose and fibrous tissue replacement in the RV was associated with lymphocytes in two (8.69%) cases. In one of these two cases, interstitial inflammatory infiltrates, predominantly lymphocytic and monocytic, diffused throughout the heart wall, associated with cardiomyocyte damage, lead to a concomitant diagnosis of lymphocytic myocarditis (Figure 5D,E).

Two (8.29%) hearts diagnosed with ACM showed pathological features of concomitant left ventricular noncompaction cardiomyopathy (LVNC), as the left ventricle presented a thin compacted epicardial layer and an endocardial noncompacted layer with prominent trabeculations and deep intertrabecular recesses (Figure 5). 

The heart weight and wall thickness of the ACM group of 23 cases were compared with a group of 15 age-matched subjects who died of SUCD from causes of death different from ACM (Non-ACM). The causes of death for the Non-ACM group were: acute myocardial infarction (five cases), myocarditis (three cases), cardiac metastasis (two cases), dilatative cardiomyopathy (one case), hypertrophic cardiomyopathy (one case), traumatic heart rupture (one case), overdose (one case), coronary artery anomalies (one case). There were no statistically significant differences between the ACM cases compared to the Non-ACM cases in regard to demographic data and BMI, heart weight, thickness of RV, LV, and IVS. Fatty-fibrous tissue in the RV, RA, LV and IVS was less in the Non-ACM group versus the ACM group and confined in limited areas. *Adipositas cordis* was diagnosed in one Non-ACM case and in none of the ACM cases.

### 3.3. Histopathological Findings in the Cardiac Conduction System

The histopathological serial sections slides of the cardiac conduction system were available for 15 (65.22%) of the 23 ACM cases. They all succumbed suddenly and unexpectedly, and the SUCD was the first manifestation of ACM. The demographic data and the cardiac conduction findings in these 15 ACM cases are presented in Table 2. 

In 12 (80%) ACM cases, the sino-atrial node (SAN) was hypoplasic due to infiltration of adipose and fibrous tissue interposed among the pacemaker cells (Figure 6). In two (13.33%) cases, a fibromuscular dysplasia of the SAN artery was detected (Figure 6A,C). In one (6.67%) case, a massive hemorrhage of the SAN and of the SAN artery was detected; in this case, abundant red blood cells were seen in the intercellular spaces, and were widely dispersed over the specialized myocardium, enclosing several working as well as specialized myocardial fibers. In three (20%) cases the SAN and its artery presented a normal structure. 

In 13 (86.67%) cases, the atrio-ventricular junctional conductive tissue (AVJ), including the atrio-ventricular node (AVN), His bundle (HB), bifurcation (BIF), left bundle branch (LBB) and right bundle branch (RBB), was hypoplastic due to infiltration by adipose and fibrous tissue (Figure 7). Disruption of the LBB due to fatty and fibrous replacement, namely a LBB block [22], was observed in two (13.33%) cases (Figure 7C).

Dispersion or septation of the AVJ, characterized by fragmentation of the AVJ within the CFC or interposition of fibrous tissue within the AVJ [20,21], was observed in seven (46.67%) cases. We observed islands of the conduction system separated from the AVN and HB and embedded in the CFC [23], or ring tissue [22], in two (13.33%) cases. Mahaim fibers, specialized connections between the AVJ and the upper interventricular septum [20,21,22], were detected in two (13.33%) cases. One Mahaim fiber was of fasciculo-ventricular or middle type and one was of bifurco-ventricular or lower type. An AVN tongue, characterized by a tongue of tissue arising from the AVN located in the central fibrous body (CFB) and directed to bridge the IVS [21], was detected in two (13.33%) cases. Duplicity, or dualism, of the HB by splitting of the HB due to interposition of fibrous tissue [23,24] was detected in one (6.67%) case.

Hypoplasia of the CFB [20,21], was observed in five (33.33%) cases. Fibromuscular dysplasia of the AVN artery [25] was observed in three (20%) cases. Cartilaginous meta-hyperplasia of the CFB [20,21,23] was observed in one (6.67%) case. A right-sided HB was detected in one (6.67%) case. 

Hemorrhage of the AVJ and infarct-like lesions, referred to cardiac massage stretching the AVJ [26], were observed in two (13.33%) cases. 

No alterations in the AVJ were detected in only one case of the 15 (equal to 6.67%) ACM cases. In all the remaining 14 (93.33%) cases, more than one alteration of the AVJ was detected simultaneously.

Table 3 shows the anatomo-morphological features listed for each of the 15 ACM SUCD cases in which the CCS has been fully analyzed. 

By comparison to these 15 ACM cases, the cardiac conduction system of the Non-ACM group of 15 age-matched subjects who died of SUCD from other causes of death was also analyzed on serial sections. Compared to the ACM group, in the Non-ACM group the amount of fatty-adipose infiltration of the CCS was significantly lower and was ≤ 5% of the extent of the conducting tissue. In particular, in the Non-ACM cases, the fatty-adipose infiltration did not cause hypoplasia of the SAN nor of the AVJ. Other findings detected in the CCS were: Islands of conduction tissue in CFB in seven (46.67%) cases, hemorrhage and infarct-like lesions in AVJ in four (26.67%) cases, LBB block by fibrosis in one (6.67%) case, AVN tongue in one (6.67%) case, SAN hypoplasia (not due to fatty-fibrous infiltration) in one (6.67%) case, AVN duplicity in one (6.67%) case, HB duplicity in one (6.67%) case, cartilaginous meta-hyperplasia of CFB in one (6.67%) case, intramural RBB in one (6.67%) case, atherosclerotic narrowing of the artery of the SAN in four (26.67%) and of the AVN artery in three (20%) cases, Mahaim fibers in five (33.33%) cases. The Mahaim fibers were: nodo-ventricular (upper type) in three cases, fasciculo-ventricular (middle type) in one case, and LBB-ventricular (lower type) in one case. Only one (6.67%) case presented no detectable alterations in the AVJ.

## 4. Discussion

Arrhythmogenic cardiomyopathy (ACM), formerly known as arrhythmogenic right ventricular cardiomyopathy (ARVC), presents clinically as ventricular arrhythmias or as sudden unexpected death without any recognized preceding cardiac dysfunction [27]. On the anatomo-pathological plane, ACM is characterized by adipose or fibro-adipose replacement, mostly of the right ventricular myocardium, but also of the left ventricle [28]. Differential diagnosis of ACM includes myocarditis, sarcoidosis, infarction of the right ventricle, dilated cardiomyopathy, Chagas disease, Brugada syndrome, idiopathic outflow tract of the right ventricle, pulmonary hypertension, and congenital heart diseases with overloaded right chambers [29]. 

### 4.1. Patients’ Characteristics

We have studied 23 ACM cases, identified in 2.07% of the total 1109 cases referred to the Università degli Studi di Milano. In our series of 823 SUCD cases, ranging in age from 22 gestational weeks to 90 years, the incidence of ACM was of 20 (2.43%) cases, considering that 3 of our ACM cases were from heart transplant. Excluding infants from these SUCD cases, our ACM prevalence of 5.22% is still lower than the 10.4% reported by Tabib et al. [30]. This lower percentage may be due to their having considered a smaller age range, of from 1 to 65 years whereas we included cases of SUCD through all ages. In a reported series by Ottaviani et al. [31] of 40 patients undergoing orthotopic heart transplantation (OHT), none of them had a diagnosis of ACM. In a larger series by Roberts et al. [32] of 314 OHT cases, ARVC was diagnosed in 1.27% of cases.

Most of our ACM cases were diagnosed at the fourth decade of life (Figure 1). ACM is usually diagnosed between 20 and 50 years of age [28]. In our series of SUCD cases encompassing all ages, the youngest victim of ACM was 5 year-old, the same youngest age reported by Tabib et al. [30]. 

In our ACM series, 16 (69.56%) patients were males and 7 (30.44%) were females. The age difference at ACM diagnosis between males (32.94 ± 14.42) and females (43.43 ± 18.36) did not differ significantly (Figure 1). A striking predominance of ACM in the male gender has been reported starting from the first very reports of this disease [11,33].

We report hereby a 34-year retrospective study of 23 consecutive cases of ACM, 20 of which presented as SUCD (Table 1). We did not find any statistically significant differences between the ACM cases compared to the Non-ACM cases in regard to demographic data and BMI.

Circumstances of death were various, but in our ACM series, 56.52% the SUCD occurred at rest; only in 5 (21.74%) cases occurred during physical effort. Athletes, in our ACM series, were 4.35% (Table 1), less than in other reported series of 31% by Corrado et al. [34]. In our series, one SCUD followed a surgery. Tabib et al. [30] reported that 9.5% of the SUCD cases happened during the perioperative period in patients without history of cardiac disease, suggesting that anesthetic drugs might trigger arrhythmias in untreated ACM patients.

### 4.2. Genetics

Mutations in the genes *plakophilin-2* (*PKP2*), *desmoplakin* (*DSP*), *desmoglein-2* (*DSG2*), *desmocollin-2* (*DSC2*), *junction plakoglobin* (*JUP*), and *transmembrane protein 43* (*TMEM43*) are strongly associated with ACM [10]. The possible ACM-associated mutations linked to conduction system impairments have been little, if at all, investigated. Moreau et al. [35] studied an ACM patient with a missense mutation (c.394C>T) in the *DSC2* gene, using a zebrafish *DSC2* model system. The *DSC2* patient-derived pluripotent stem cells were reprogrammed and differentiated into cardiomyocytes, the human-induced pluripotent stem cells cardiomyocytes (hiPSC-CM). Based on the responsiveness to antiarrhythmic drugs, a short QT interval at low heart rate was associated with *DSC2* ACM. Recently, Hayashi et al. [36] performed Whole-exome sequencing (WES) of 23 probands diagnosed with early-onset (<65 years) of cardiac conduction system diseases analyzing 117 genes linked to arrhythmogenic diseases or cardiomyopathies. Cellular electrophysiological study and in vivo zebrafish cardiac assay showed that two variants in *Potassium Voltage-Gated Channel Subfamily H Member 2* (*KCNH2)* and *Sodium Voltage-Gated Channel Alpha Subunit 5* (*SCN5A*), four variants in *Sodium Voltage-Gated Channel Alpha Subunit 10* (*SCN10A*), and one variant in *Myosin Heavy Chain 6* (*MYH6*) genes resulted from “uncertain significance” to “likely pathogenic” in six probands.

In our series, post-mortem genetic tests were provided by the referring center in only one (4.35%) case and demonstrated a pathogenic mutation in gene *PKP2*, which is consistent with the autopsy diagnosis of ACM, being a common abnormal gene for ACM [1,7]. It has been reported by Corrado et al. [7] that 50% of the affected patients have a positive family history. In our series, only 17.4% of the subjects had a familial occurrence of SUCD < 50 years and 4.35% had a familial occurrence of ACM. This lower occurrence in our cases can be related to the fact that the clinical information accompanying our cases was scarce, despite the efforts by the first author to gather more data. The fact that only one subject out of 23 ACM cases sent by the referring centers was accompanied by complete genetic testing is certainly a limitation of our study. New policies should be adopted to ensure that all hearts that are analyzed must be accompanied by satisfactory family and patient’s history. A solid collaboration with geneticists working on each newly submitted case should also be established.

### 4.3. Anatomo-Pathological Findings of the Heart

Grossly, all our 23 ACM hearts showed global dilatation of the RV cavity, presenting bulging and/or aneurysms, with transmural muscle loss and thinning of the RV wall (Figure 2), confirmed by transillumination (Figure 3). The heart weight mean ± SD, 410.33 ± 120.67 g, was generally increased.

Histopathologically, in all (100%) of our 23 cases of ACM, decreased thickness of the RV wall, transmural myocardial cell loss with adipose and fibrous tissue replacement in the RV, in absence of significant coronary artery disease or other known cardiac causes, were observed. The adipose tissue was either disorganized or organized in columns (Figure 4). Our findings are consistent with the typical anatomo-pathological findings of ACM [15,16,28].

Five (21.74%) patients had isolated RV involvement (Figure 2D,E); 18 (78.26%) cases had LV involvement (Figure 5), and 13 (56.52%) cases had IVS involvement. Fibro-fatty infiltration was also detected in the RA in 20 (86.96%) cases (Figure 4E,F) and in the LA in 10 (43.48%) cases. It is now well established that ACM broadly includes biventricular and left-dominant phenotypic forms [37], but little, if anything, is known about the involvement of atria. 

Concomitant hypertrophic cardiomyopathy was detected in three (13.04%) cases and LVNC in two (8.29%) cases (Figure 5). Further studies should focus on possible ACM variants showing overlapping phenotypes with other cardiomyopathies. 

In our 23 ACM cases, the adipose and fibrous tissue replacement in the RV was associated with lymphocytes in two (8.69%) cases. Our findings are similar to those reported by Tabib et al. [30] that reported isolated lymphocytes in 5.5% of cases. In one (4.35%) of our cases a concomitant diagnosis of lymphocytic myocarditis was established (Figure 5D,E). Myocarditis has been reported to be associated to ACM, as a primary event or as a reaction to spontaneous cell death [38]. Cardiac sarcoidosis can mimic ACM or can be associated to ACM [39,40].

We did not find any statistically significant differences between the ACM cases compared to the Non-ACM cases in regard to heart weight, thickness of RV, LV, and IVS.

The adipose and fibrotic replacement in ACM patients’ hearts is not associated with cardiomyocyte transdifferentiation, but the cells responsible for these processes are stromal fibroblastoid cells. Lombardi et al. [41] reported that, in ACM, a subset of fibro-adipogenic progenitors (FAPs) express desmosome proteins and differentiate to adipocyte upon deletion of the *DSP* gene. Cardiac mesenchymal stromal cells (C-MSC) are the most abundant cells in the heart, with propensity to differentiate into several cell types, including adipocytes; they contribute to the adipogenic substitution of cardiomyocytes observed in ACM [42]. C-MSC are a source of myofibroblasts and participate in ACM fibrotic remodeling, being highly responsive to ACM-characteristic excess of transforming growth factor-β (TGF-β) [43].

### 4.4. Histopathological Findings in the Cardiac Conduction System

The present study evaluated the anatomo-pathological profile of ACM with special reference to the disease involvement of the cardiac conduction system, by studying the conducting tissue collected from whole hearts. This study revealed a hypoplasia due to fatty-fibrous involvement of the sino-atrial node (SAN) in 80% of cases, and of the atrio-ventricular junction (AVJ), atrio-ventricular junction (AVJ) including the atrio-ventricular node, the His bundle, the bifurcation, the left bundle branch and the right bundle branch in 86.67% of cases (Table 2) (Figure 6 and Figure 7). Tabib et al. [30] reported a lower percentage of fibro-fatty infiltration, detected in 68% of cases, but they considered only the HB and not the entire AVJ, as we did in our study. This fatty-fibrous involvement of the conducting tissue might be the underlying morphological background for the development of malignant arrhythmias leading to SUCD. 

To the best of our knowledge, this is the first report of a systematic histopathological analysis of the cardiac conduction system carried out on serial sections in ACM cases. In fact, Tabib et al. [30] studied histopathologically only the His bundle and the bundle branches. Peters [44] studied the conduction system only through the evaluation of ECG, concluding that the fibro-fatty lesions in the conduction system were present in only 6% of cases, suggesting that the Tabib et al.’s findings [30] of more than 60% fibrotic lesions in the conduction system were rather secondary to hypertension. On the contrary, hypertension was absent from our cases. Our findings differ with the conclusion of Peters [44] that ACM usually spares the conduction system. As the name implies, arrhythmias are the hallmark of ACM, not only from the fatty-fibrous disruption of the ventricular myocardium that accounts for reentrant ventricular tachycardia [22,28], but also from the fatty-fibrous involvement of the conduction system itself. Several abnormalities of the cardiac conduction system have been detected in our asymptomatic ACM individuals who succumbed suddenly and unexpectedly: Hypoplasia of SAN and/or AVJ due to fatty-fibrous involvement (Figure 6 and Figure 7), AVJ dispersion and/or septation, central fibrous body (CFB) hypoplasia, fibromuscular dysplasia of CCS arteries, hemorrhage and infarct-like lesions of CCS, islands of conduction tissue in CFB, Mahaim fibers, LBB block by fibrosis, AVN tongue, HB duplicity, cartilaginous meta-hyperplasia, and right sided HB (Table 2). All these anomalies, were also detected in Non-ACM subjects, except the fatty-fibrous involvement and the fibromuscular dysplasia of the CCS that have been detected only in the ACM individuals. Thiene et al. [28] reported CCS anomalies in 10.5% of ACM cases. They did not describe the CCS findings, but stated that the CCS is usually spared by the disease. On the contrary, the CCS anomalies detected in our cases might represent the morphological basis for arrhythmogenic death. 

Bharati and Lev [45] histopathologically examined the CCS in 14 athlete victims of SUCD. They identified, as congenital anomalies, duplicity of the SAN and AVJ accessory pathways. The acquired abnormalities included frequent mononuclear cell infiltration SAN, fat and fibrosis to a varying degree in all parts of the conduction system, and focal fibrotic scar areas in the ventricular septum. They did not classify any of their cases as ACM victims. Despite that our series is different from the one analyzed by Bharati and Lev [45], we are hereby reaching the same conclusion that it is evident that these findings were present for a long period of time while the subjects were asymptomatic. The fatty-fibrous infiltration frequently detected in ACM represents the morphological substrate for the development of arrhythmic reentry mechanisms, eventually leading to ventricular fibrillation and SUCD. The fatty-fibrous involvement of the CCS has been observed starting from the age of five years in our series, but it is still unknown if it is congenital in nature, i.e., present from birth. The hypothesis that the fatty-fibrous involvement of the CCS is congenital in nature needs to be further investigated and proven. Bharati and Lev [45] stated that the fatty-fibrous anomalies of the cardiac conduction system in young athletes that died suddenly were acquired in nature. On the contrary, we suggest that they are congenital in nature and associated to specific genetic defects underlying ACM responsible for vulnerable conduction systems. In our ACM cases, the CCS findings were similar to the findings of Burke and Virmani [46] detected in the CCS of athletes who died suddenly: premature sclerosis of the CCS, nonatherosclerotic dysplasia of the AVN artery, and accessory pathways. 

It is now well known that the CCS cells harbor both mechanical anchoring junctions and electrical gap junction and ion channels. Cell junctions in the specialized cardiomyocytes of the CCS have been studied in in murine models suggesting a potential functional role of anchoring junction components in the CCS. Further studies on the role of junctions specifically within the CCS in the mouse and human will provide insight into uncovering the mechanisms underlying ACM and other human cardiac diseases associated with cell – cell junction defects [47].

The ECG findings in our series (Table 1) are consistent with the patterns previously reported by Corrado et al. [7]. Brugada syndrome, as confirmed in one (4.35%) of our cases, has been already reported by Corrado et al. [48] in confirmed cases of ARVC, meanwhile named ACM. Young subjects affected by ACM diagnosed by ECG abnormalities, confirmed by echocardiography, cardiac magnetic resonance and, if needed, also endomyocardial biopsy, would be advised of lifesaving sport disqualification. However, biventricular of LV dominant ACM can still escape ECG screening [1,4]. If the ACM asymptomatic patients who succumbed suddenly and unexpectedly would have been diagnosed in life, they could have been treated with the various treatments available, including implantable defibrillator and heart transplant [1]. 

The fact that ACM is related to lethal arrhythmias or heart block due to abnormalities of the conduction system should gather a renewed interest in investigating the CCS on serial sections. Nowadays, despite the progress of molecular autopsy, there are always cases of SUCD whose post-mortem investigation was unable to identify a clear cause of death [49].

Our findings provide evidence in support of a fatty-fibrous involvement of the CCS in ACM, suggesting that a fatal electrical instability of the heart may be the final common pathway for sudden death. However, the nature of the changes in the CCS and its relation to ACM is not yet fully explained. The prevention of ACM and arresting its silent progression to fatal cardiac events are yet to be addressed. Postnatal morphogenesis of the CCS, represented by AVJ molding and shaping through cell death and replacing in an orderly programmed way, is an important part of its normal development. Areas of resorptive degeneration, signs of peri-natal shaping and molding of the atrio-ventricular junctional conducting tissue in infants and term fetuses [21,23], were not detected in our ACM and Non-ACM cases. Resorptive degeneration, if defective, could leave in place some accessory communication between the atrio-ventricular pathway and the adjacent ordinary myocardium [50], as detected in our cases of SUCD. Accessory pathways, islands of CCS in the CFB, AVN tongue, and duplicity of AVJ structures cannot be considered specifically related to ACM, having been detected also in other forms of SUCD [4,18,21,23,24,50] and, in the current study, in the ACM similarly to the Non-ACM cases. However, according to James [51], a defective apoptosis could play a role in the pathogenesis ACM and favoring lethal electrical instability simultaneously in the CCS and common working myocardium. 

Our findings provide evidence in support of a conduction system involvement for ACM, suggesting that the final common pathway could be some form of fatal cardiac electrical instability of the heart. However, the nature of the pre- or post-natal changes in the cardiac conduction system and its relation to ACM is not yet fully explained.

Our study of the CCS on serial sections has been carried out *post-mortem*. Additional clinico-pathological and imaging correlations of the CCS findings would allow for intervention *intra-vitam* by identifying subjects at risk for sudden death. Subsequent actions to avoid the lethal arrhythmic event, such as disqualification from athletic activities and implantation of a portable defibrillator, would be foreseen. Are there reliable diagnostic techniques able to visualize the CCS *intra-vitam*? Recently, Kawashima et al. [52] carried out a specialized physical three-dimensional (3D) computed tomography (CT) micro-dissection of serial sections imaging of the CCS in the human body. The technique identified that when the cardiac inclination changed from standing to lying, the SAN shifted from the dorso-superior to the right outer position and the AV conduction axis changed from a vertical to a leftward horizontal position. In situ localization of the human CCS provided accurate anatomical localization with morphometric data and useful correlation between heart inclination and CCS rotation axes for predicting variable in the CCS in human living body. 

Our histopathological CCS findings will be useful to be paired with advanced future imaging modalities and methodology for further accurate prediction of subjects at risk for sudden death, suffering from ACM or other cardiomyopathies.

As general, forensic and even molecular autopsies may be inconclusive in evaluating cases of sudden unexpected death, an accurate post-mortem examination in all cases of suspected ACM, including in particular the examination of the cardiac conduction system on serial sections, is useful to identify the final events leading to death. 

### 4.5. Limitations

Since the hearts were collected mostly from forensic autopsies already performed by the referring centers, it was not possible to obtain a complete clinical record or the genetic tests of the family members. 

A limitation of this study is the lack of genetic data, available only for one subject of our 23 ACM series. 

Since the Lino Rossi Research Center of the Università degli Studi di Milano, Milan, Italy mostly receives cases of sudden unexpected death, our control group consisted of Non-ACM subjects who died suddenly and unexpectedly for causes different from ACM (Non-ACM group). Therefore, the percentage of SUCD cases in our series might have been affected by selection bias.

## 5. Conclusions

To our knowledge, this is the first original article that performed a serial evaluation of the histopathological findings in the cardiac conduction system of asymptomatic individuals who succumbed suddenly and unexpectedly from ACM. As the name implies, arrhythmias are the hallmark of ACM, not only from the fatty-fibrous disruption of the ventricular myocardium that accounts for reentrant ventricular tachycardia [22,28], but also from the fatty-fibrous involvement of the conduction system itself. The careful examination of the cardiac conduction system on serial sections documented the adipose and fibrous infiltration of the cardiac conduction system in ACM. Future research should focus on application of this knowledge on CCS anomalies to be added to diagnostic criteria or at least to be useful to detect the patients with higher sudden death risks. 

## Figures and Tables

**Figure 1 diagnostics-11-01323-f001:**
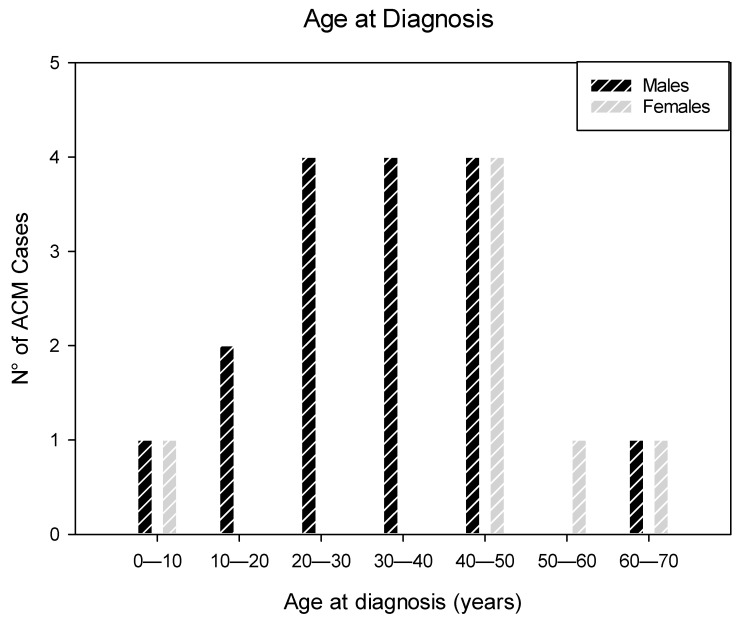
This graph shows the gender and age frequency distribution of the diagnosis of the 23 cases of arrhythmogenic cardiomyopathy (ACM). ACM was more frequent in males (69.56%) than in females (30.44%), ranging in age from 5 to 65 (mean age ± SD, 36.13 ± 16.06) years. The age difference at ACM diagnosis between males (32.94 ± 14.42) and females (43.43 ± 18.36) was not statistically significant (*p* = 0.154).

**Figure 2 diagnostics-11-01323-f002:**
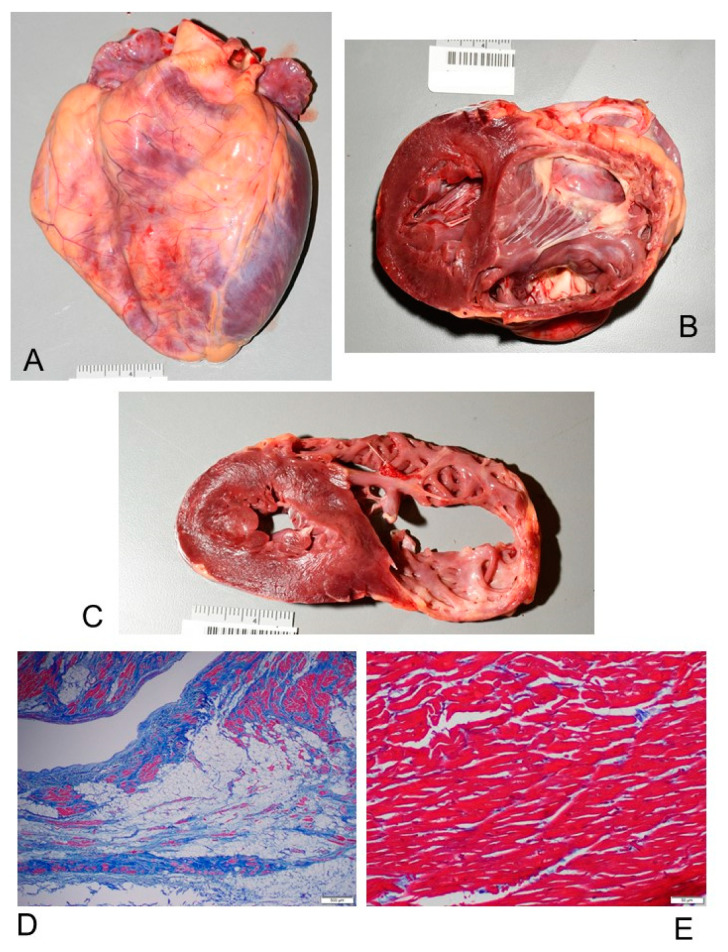
Gross examination of the heart shows global dilatation, with transmural muscle loss and thinning of the right ventricle (RV). At the bottom there is a bar scale showing millimeters (mm) and centimeters (cm). Case # 23, 28-year-old man who died suddenly and unexpectedly while playing sport; the heart weight was 450 g. A diagnosis of arrhythmogenic cardiomyopathy (ACM) was established at post-mortem examination. (**A**) Gross aspect of the heart shows global dilatation; anterior view. (**B**) Opened heart by a biventricular section shows severe thinning of RV, which presents a saccular aneurysm of the RV chamber toward cardiac apex. (**C**) Biventricular section shows the contrast between the normal left ventricle (LV) and interventricular septum compared to the thinned RV. (**D**) Histopathological slide of the RV wall shows extensive replacement of myocytes (stained red) by fibrous scar (stained blue) and adipose tissue. Trichromic Heidenhain (Azan), 100×. (**E**) In contrast, the histopathological slide of the LV shows the normal presence of the myocytes. Trichromic Heidenhain (Azan), 100×.

**Figure 3 diagnostics-11-01323-f003:**
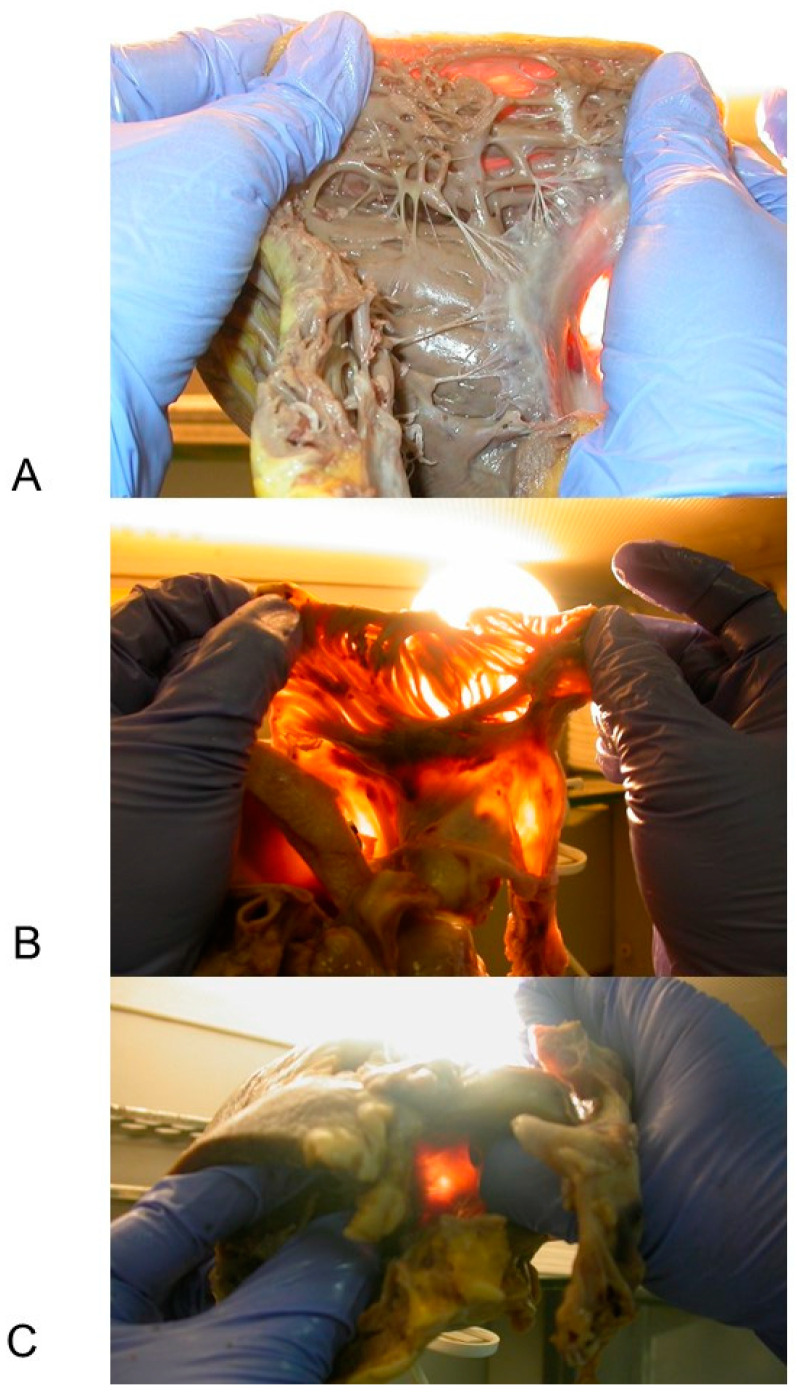
Gross features of heart’s by transillumination: The opened heart was placed against a source of light. Case # 21, 47-year-old man who died suddenly and unexpectedly; the heart weight was 415 g. A diagnosis of arrhythmogenic cardiomyopathy (ACM) was established at post-mortem examination. The histopathological findings of this case are shown in Figure 5. (**A**) The right ventricle (RV) wall appears translucent due to transmural muscle loss of the RV wall. (**B**) The right atrium wall is so thin to appear devoid of muscle at transillumination. (**C**) The *pars membranacea septi*, reference point for the heart dissection to isolate the heart specimen containing the atrio-ventricular junction, has a significantly increased translucency.

**Figure 4 diagnostics-11-01323-f004:**
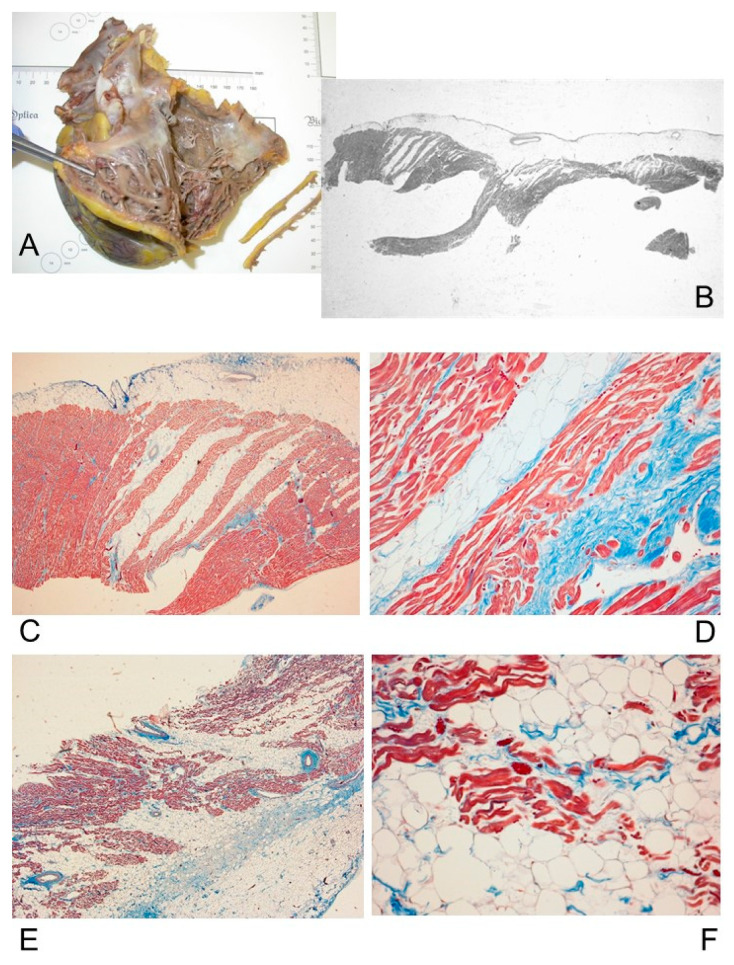
Case # 18, 31-year-old man who died suddenly and unexpectedly; the heart weight was 419 g. A diagnosis of arrhythmogenic cardiomyopathy (ACM) was established at post-mortem examination. (**A**) Gross examination of the opened heart shows thinning of the right ventricle (RV). The RV wall had a thickness of 3 mm. (**B**–**D**) Antero-lateral histopathological section of the wall of the right ventricle (RV), with epicardium on top and endocardium on bottom, largely characterized by the replacement of its cardiomyocytes mainly by fat and also fibrous tissue. Trichromic Heidenhain (Azan); (**B**) 5×; (**C**) 40×; (**D**) 200×. (**E**,**F**) Antero-lateral histopathological section of the wall of the right atrium: areas of fibro-fatty replacement. Azan, 40×; 200×.

**Figure 5 diagnostics-11-01323-f005:**
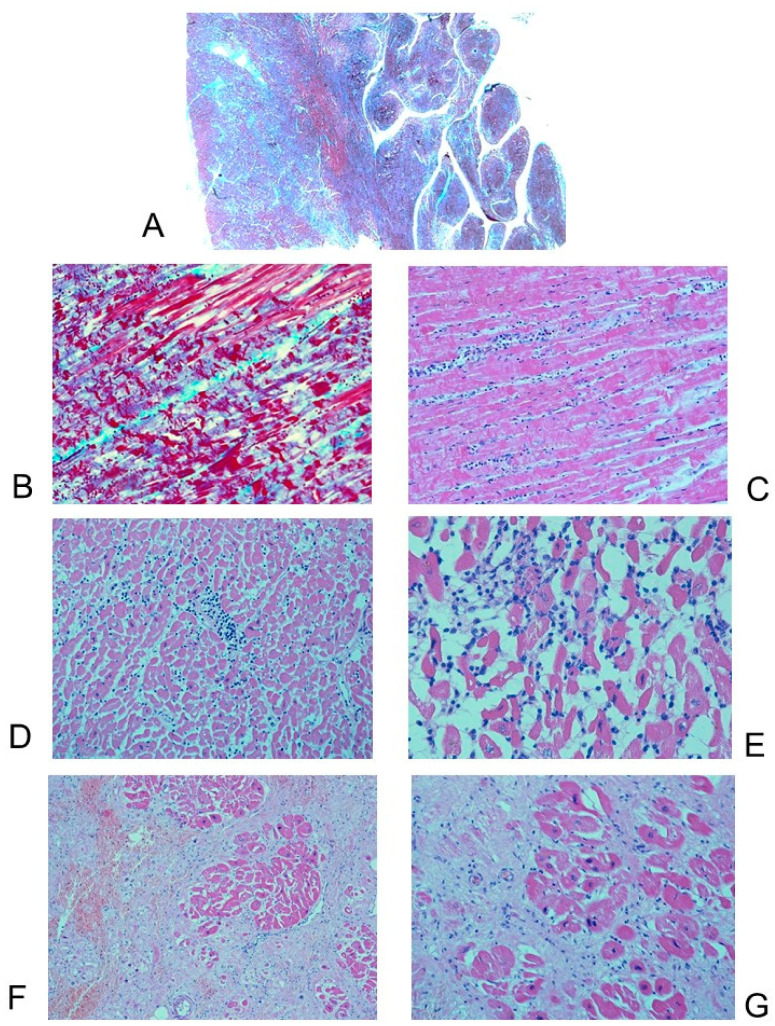
Histopathological sections of the left ventricle (LV) wall. Case # 21, 47-year-old man who died suddenly and unexpectedly. Same case of Figure 3. (**A**) LV, anterior wall: Thin compacted epicardial layer and an endocardial noncompacted layer with prominent trabeculations and deep intertrabecular recesses, as features of ventricular noncompaction cardiomyopathy (LVNC). Trichromic Heidenhain (Azan), 5×. (**B**,**C**) Close-up views of (**A**) disclose anisoinotropism with contraction bands necrosis, suggestive for acute myocardial infarction. (**B**) Azan, 400×; (**C**) Hematoxylin–Eosin (HE), 400×. (**D**,**E**) LV, lateral wall: Interstitial inflammatory infiltrates, predominantly lymphocytic and monocytic, associated with cardiomyocyte damage, leading to a concomitant diagnosis of lymphocytic myocarditis. HE; (**D**) 200×; (**E**) 400×. (**F**,**G**) LV, lateral wall: extensive fibrous and also fatty myocardial replacement, embedding residual myocytes. HE; (**F**) 200×; (**G**) 400×.

**Figure 6 diagnostics-11-01323-f006:**
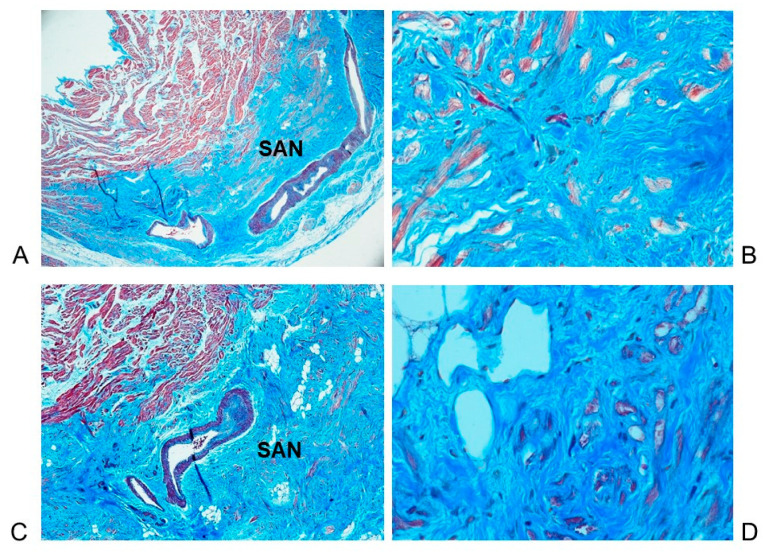
(**A**,**C**): Two histopathological serial sections of the sino-atrial node. The SAN is centered by the SAN node artery which presents a narrowing due to fibromuscular dysplasia. (**B**,**D**) At higher magnification, the sino-atrial node (SAN) is hypoplasic due to infiltration of adipose and fibrous tissue, interposed among the pacemaker cells. Case # 22, 50-year-old man who died suddenly and unexpectedly. Stain: Trichromic Heidenhain (Azan). Magnification: (**A**,**C**) 20×; (**B**,**D**) 400×.

**Figure 7 diagnostics-11-01323-f007:**
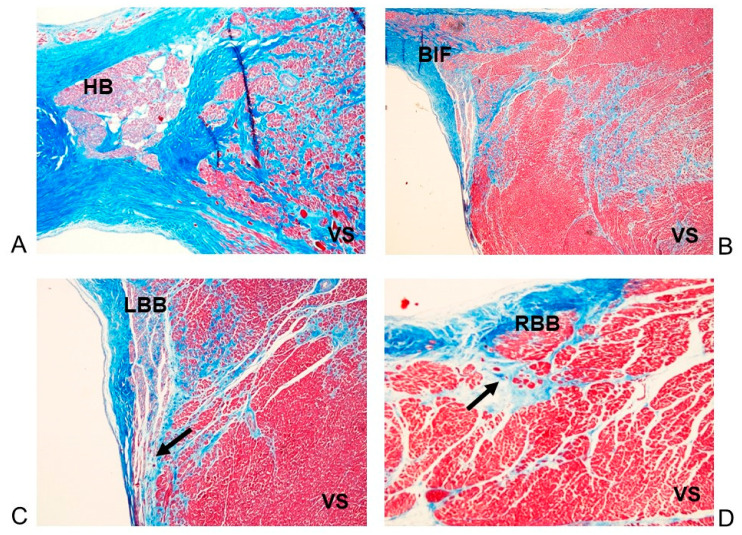
(**A**–**D**) Four histopathological serial sections of the atrio-ventricular junctional conductive tissue (AVJ). The His bundle (HB) (**A**), bifurcation (BIF) (**B**), left bundle branch (LBB) (**C**) and right bundle branch (RBB) (**D**) are hypoplasic due to infiltration of adipose and fibrous tissue, interposed among the specialized cells. Note the myocardial replacement by fibrous tissue in the interventricular septum (IVS), as part of the arrhythmogenic cardiomyopathy (ACM). Arrows point to the bundle branches. Case # 19, 51-year-old woman who died suddenly and unexpectedly. Stain: Trichromic Heidenhain (Azan); Magnification: (**A**,**C**) 20×; (**B**,**D**) 40×.

**Table 1 diagnostics-11-01323-t001:** Demographic and clinical data in 23 ACM consecutive cases.

	ACM (*N* = 23)
Gender (M/F)	16 (69.56%)/7 (30.44%)
Mean age ± SD (years)	36.13 ± 16.06; range: 5–65
Ethnicity (W/B)	22 (95.65%)/1 (4.35%)
Familial occurrence of SUCD < 50 years	4 (17.4%)
Familial occurrence of ACM	1 (4.35%)
Athlete	1 (4.35%)
Previous symptoms	5 (21.74%)
Diagnosis of ACM (IV/PM)	3 (13.04%)/20 (86.96%)
ECG	6 (26.08%)
Low QRS voltages	2 (8.69%)
Epsilon waves	1 (4.35%)
Incomplete right bundle block, Brugada syndrome	1 (4.35%)
Ventricular fibrillation	2 (8.69%)
Mode of death or failure:	
SUCD	20 (86.96%)
at rest	13 (56.52%)
on effort	5 (21.74%)
unknown	2 (8.69%)
CHF requiring OHT	3 (13.04%)

ACM = arrhythmogenic cardiomyopathy; M = male; F = female; W = white; B = black; IV = intra-vitam; PM = post-mortem; ECG = electrocardiogram; SUCD = sudden unexpected cardiac death; CHF = congestive heart failure; OHT= orthotopic heart transplant.

**Table 2 diagnostics-11-01323-t002:** Demographic data and cardiac conduction histopathological findings in 15 ACM cases who succumbed suddenly and unexpectedly.

	ACM (*N* = 15)
Gender (M/F)	11 (77.33%)/4 (26.67%)
Mean age ± SD (years)	35.07 ± 13.85; range: 5–51
Ethnicity (W/B)	14 (93.33%)/1 (6.67%)
Familial occurrence of SUCD < 50 years	3 (20%)
Familial occurrence of ACM	1 (6.67%)
AVJ hypoplasia due to fatty-fibrous involvement	13 (86.67%)
SAN hypoplasia due to fatty-fibrous involvement	12 (80%)
AVJ dispersion and/or septation	7 (46.67%)
CFB hypoplasia	5 (33.33%)
Fibromuscular Dysplasia of AVN artery	3 (20%)
Fibromuscular Dysplasia of SAN artery	2 (13.33%)
Islands of conduction tissue in CFB	2 (13.33%)
Hemorrhage and infarct-like lesions in AVJ	2 (13.33%)
LBB block by fibrosis	2 (13.33%)
AVN tongue	2 (13.33%)
Mahaim fiber	2 (13.33%)
Hemorrhage in SAN	1 (6.67%)
HB duplicity	1 (6.67%)
Cartilaginous meta-hyperplasia of CFB	1 (6.67%)
Right sided HB	1 (6.67%)

ACM = arrhythmogenic cardiomyopathy; M = male; F = female; W = white; B = black; IV = intra-vitam; PM = post-mortem; SUCD = sudden unexpected cardiac death; SAN =sino-atrial node; HB = His bundle; AVN = atrio-ventricular node; AVJ = atrio-ventricular junction; CFB = central fibrous body; LBB = left bundle branch.

**Table 3 diagnostics-11-01323-t003:** Demographic data and anatomo-pathological findings of the heart listed for each of the 15 ACM patients who succumbed suddenly and unexpectedly.

*N*	Age (Years)	Sex	Heart Weight (g)	RV Bulging/Aneurysm	RV Wall Thickness (mm)	Patchy Inflammation	LV Involved	Cardiac Conduction System
1	17	M	430	Yes	4	No	Yes	SAN and AVJ fatty-fibrous involvement; right sided HB
2	17	M	510	No	5	No	No	SAN and AVJ fatty-fibrous involvement; AVJ septation
3	29	M	408	No	5	No	No	SAN and AVJ fatty-fibrous involvement; islands of conduction tissue in CFB; hemorrhage and infarct-like lesions in AVJ; LBB block by fibrosis
4	40	M	508	No	4.5	No	No	SAN and AVJ fatty-fibrous involvement
5	33	M	500	Yes	2	No	Yes	SAN and AVJ fatty-fibrous involvement; AVJ dispersion
6	5	F	92	Yes	2	Yes	No	SAN fatty-fibrous involvement; septated HB; CFB hypoplasia; Mahaim fiber LBB-ventricular (lower type); Cartilaginous meta-hyperplasia of CFB
7	44	M	664	Yes	5	Yes	Yes	SAN and AVJ fatty-fibrous involvement; septated HB; CFB hypoplasia
8	46	F	377	No	8	No	Yes	AVJ fatty-fibrous involvement; AVJ dispersion; SAN artery fibromuscular dysplasia
9	47	F	371	Yes	5	No	Yes	Hemorrhage in SAN; AVJ fatty-fibrous involvement; fibromuscular dysplasia of AVN artery; Mahaim fiber fasciculo-ventricular (middle type);
10	31	M	419	Yes	3	No	Yes	SAN and AVJ fatty-fibrous involvement; septated AVN and HB; CFB hypoplasia; fibromuscular dysplasia of AVN artery; islands of conduction tissue in CFB; HB duplicity
11	51	F	288	No	8	No	Yes	SAN and AVJ fatty-fibrous involvement; AVJ dispersion; LBB block by fibrosis
12	41	M	391	Yes	5	No	Yes	SAN and AVJ fatty-fibrous involvement; fibromuscular dysplasia of AVN artery
13	47	M	415	Yes	5	Yes	No	SAN and AVJ fatty-fibrous involvement; hemorrhage and infarct-like lesions in AVJ
14	50	M	440	Yes	6	No	No	SAN and AVJ fatty-fibrous involvement; CFB hypoplasia; SAN artery fibromuscular dysplasia
15	28	M	450	Yes	4	No	No	Unremarkable

ACM = arrhythmogenic cardiomyopathy; M = male; F = female; g, grams; mm, millimeters; RV, right ventricle; SAN = sino-atrial node; HB = His bundle; AVN = atrio-ventricular node; AVJ = atrio-ventricular junction; CFB = central fibrous body; LBB = left bundle branch.

## Data Availability

Anonymized study data are available on request from the corresponding author. The data are not publicly available due to the privacy and confidentiality in personal data collection and processing, in accordance with EU and international legislation.

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
