# Peer review of "Sudden Unexpected Death Associated with Arrhythmogenic Cardiomyopathy: Study of the Cardiac Conduction System"

_diagnostics, 2021, doi:10.3390/diagnostics11081323_

Round 1

Reviewer 1 Report

The authors performed a retrospective study on cardiac tissue samples from individuals who died of sudden death and affected by arrhythmogenic cardiomyopathy (ACM) to examine the presence of fibro-fatty infiltration in the cardiac conduction system (CCS). 

This is the first complete analysis of the CCS of ACM patients, proposing the involvement of CCS in arrhythmia occurrence, besides the well-accepted presence of ventricular tachycardias arising from the right ventricle.

The potential relevance of this study needs to be strengthened in line with the following comments:

  • There are still no definite criteria to diagnose ACM at autopsy. Moreover, other diseases are in differential diagnosis. To further confirm to have included in the analysis the correct cohort of patients, it would be useful to perform additional analyses (e.g. apoptosis/necrosis assay, quantification of residual myocytes that should be <60% according to the present literature, hypertrophy of cardiomyocytes). Please, also add in the Introduction the literature about autopsy investigations in ACM and this limitation.
  • A table in which each ACM patient is listed with all the specific features measured (e.g. heart weight, wall thickness, bulging, aneurysms, areas of gross infiltrate, type of infiltrate, etc) could be useful. In this way, it could be possible also to correlate if the alterations in the CCS are present only in individuals with ventricular fibro-fatty substitution or not.
  • Table 1: please specify that ECG data are about only 6 patients out of 23 and recalculate the percentages.
  • Paragraph 3.2: please split into separate paragraphs the macroscopic and microscopic findings.
  • In the Discussion, an overview of ACM-associated mutations that have been linked to conduction system impairments could be useful. The authors could speculate and comment about it.
  • To strengthen the findings and complete the discussion with also evidence in murine models, please revise Cell Junctions in the Specialized Conduction System of the Heart 3109/15419061.2014.905928.
  • The adipose and fibrotic replacement in ACM patients’ hearts is not associated with cardiomyocyte transdifferentiation, but the cells responsible for these processes are stromal fibroblastoid cells (doi: 10.1093/eurheartj/ehv579; doi: 10.1161/CIRCRESAHA.115.308136; doi: 10.3390/ijms22052673). In the Discussion, please contextualize your findings taking into account the origin of the fibro-fatty tissue.
  • Conclusions: the final message of your work can be better explain. It could be interesting to understand if it could be added to diagnostic criteria or at least to be useful to detect the patients with higher sudden death risks. Are there reliable diagnostic techniques able to visualize the CCS? (For example revise: first in situ 3D visualization of the human cardiac conduction system and its transformation associated with heart contour and inclination 10.1038/s41598-021-88109-7).
  • Graphical abstract is missing.
  • Please, improve the quality of the images.
  • Please, check the cited literature and modify it when is not appropriately referred to the text. 

Author Response

July 8, 2021

Re: Sudden Unexpected Death Associated with Arrhythmogenic Cardiomyopathy:

Study of the Cardiac Conduction System  (Ms. ID: diagnostics-1271816)

Authors’ Responses to REVIEWERS’ Remarks

We thank the Editor-in-Chief and the two Reviewers for their constructive comments.  Replies to individual comments are stated below each comment. The reviewer’s comments are shown in bold font and our responses are shown in italics.

TO REVIEWER # 1

  1. There are still no definite criteria to diagnose ACM at autopsy. Moreover, other diseases are in differential diagnosis. To further confirm to have included in the analysis the correct cohort of patients, it would be useful to perform additional analyses (e.g. apoptosis/necrosis assay, quantification of residual myocytes that should be <60% according to the present literature, hypertrophy of cardiomyocytes). Please, also add in the Introduction the literature about autopsy investigations in ACM and this limitation.

Response: Thank you for this helpful comment. In response to the Reviewer’s suggestion, the following paragraph has been added to the manuscript:

“Autopsy investigation has its limitations as well. For example, there are still no definite criteria to diagnose ACM and there is still some confusion about the use of the term ACM rather than ARVC and if the diagnostic criteria at autopsy refer to which one of the two or to both. Moreover, other diseases are in differential diagnosis with ACM. To confirm the correct diagnosis, it is useful to evaluate the presence of apoptosis, to quantify the residual myocytes that should be <60%, and to evaluate the hypertrophy of cardiomyocytes [15,16]”.

[Introduction, page 3, lines 92-97]

The following inherent references have been added to the manuscript:

“15.       Marcus, F.I.; McKenna, W.J.; Sherrill, D.; Basso, C.; Bauce, B.; Bluemke, D.A.; Calkins, H.; Corrado, D.; Cox, M.G.P.J.; Daubert, J.P.; et al. Diagnosis of arrhythmogenic right ventricular cardiomyopathy/Dysplasia: Proposed modification of the task force criteria. Circulation 2010, 121, 1533–1541, doi:10.1161/CIRCULATIONAHA.108.840827.

  1. Corrado, D.; Van Tintelen, P.J.; McKenna, W.J.; Hauer, R.N.W.; Anastastakis, A.; Asimaki, A.; Basso, C.; Bauce, B.; Brunckhorst, C.; Bucciarelli-Ducci, C.; et al. Arrhythmogenic right ventricular cardiomyopathy: Evaluation of the current diagnostic criteria and differential diagnosis. Eur. Heart J. 2020, 41, 1414–1427, doi:10.1093/eurheartj/ehz669.

[References, page 25, lines 739-746]

  1. A table in which each ACM patient is listed with all the specific features measured (e.g. heart weight, wall thickness, bulging, aneurysms, areas of gross infiltrate, type of infiltrate, etc) could be useful. In this way, it could be possible also to correlate if the alterations in the CCS are present only in individuals with ventricular fibro-fatty substitution or not.

Response. Thank you for this useful comment. According to the reviewer’s suggestion, the following Table in which each ACM patient who underwent a detailed conducting tissue investigation is listed with all the cardiac anatomo-pathological findings displayed:

Table 3. Demographic data and anatomo-pathological findngs of the heart listed for each of the 15 ACM patients who succumbed suddenly and unexpectedly.

Age (years)

Sex

Heart weight (g)

RV bulging/ aneurysm

RV wall thickness (mm)

Patchy inflammation

LV involved

Cardiac Conduction System

1

17

M

430

Yes

4

No

Yes

SAN and AVJ fatty-fibrous involvement; right sided HB

2

17

M

510

No

5

No

No

SAN and AVJ fatty-fibrous involvement; AVJ septation

3

29

M

408

No

5

No

No

SAN and AVJ fatty-fibrous involvement; islands of conduction tissue in CFB; hemorrhage and infarct-like lesions in AVJ; LBB block by fibrosis

4

40

M

508

No

4.5

No

No

SAN and AVJ fatty-fibrous involvement

5

33

M

500

Yes

2

No

Yes

SAN and AVJ fatty-fibrous involvement; AVJ dispersion

6

5

F

92

Yes

2

Yes

No

SAN fatty-fibrous involvement; septated HB; CFB hypoplasia; Mahaim fiber LBB-ventricular (lower type); Cartilaginous meta-hyperplasia of CFB

7

44

M

664

Yes

5

Yes

Yes

SAN and AVJ fatty-fibrous involvement; septated HB; CFB hypoplasia

8

46

F

377

No

8

No

Yes

AVJ fatty-fibrous involvement; AVJ dispersion; SAN artery fibromuscular dysplasia

9

47

F

371

Yes

5

No

Yes

Hemorrhage in SAN; AVJ fatty-fibrous involvement; fibromuscular dysplasia of AVN artery; Mahaim fiber fasciculo-ventricular (middle type);

10

31

M

419

Yes

3

No

Yes

SAN and AVJ fatty-fibrous involvement; septated AVN and HB; CFB hypoplasia; fibromuscular dysplasia of AVN artery; islands of conduction tissue in CFB; HB duplicity

11

51

F

288

No

8

No

Yes

SAN and AVJ fatty-fibrous involvement; AVJ dispersion; LBB block by fibrosis

12

41

M

391

Yes

5

No

Yes

SAN and AVJ fatty-fibrous involvement; fibromuscular dysplasia of AVN artery

13

47

M

415

Yes

5

Yes

No

SAN and AVJ fatty-fibrous involvement; hemorrhage and infarct-like lesions in AVJ

14

50

M

440

Yes

6

No

No

SAN and AVJ fatty-fibrous involvement; CFB hypoplasia; SAN artery fibromuscular dysplasia

15

28

M

450

Yes

4

No

No

Unremarkable

ACM = arrhythmogenic cardiomyopathy; M = male; F = female; g, grams; mm, millimeters; RV, right ventricle; SAN =sino-atrial node; HB = His bundle; AVN = atrio-ventricular node; AVJ = atrio-ventricular junction; CFB = central fibrous body; LBB = left bundle branch.

[Results, Table 3, page 17, lines 413-418]

  1. Table 1: please specify that ECG data are about only 6 patients out of 23 and recalculate the percentages.

Response:  According to the reviewer’s comment, the total number of patients with available ECG data and their percentage have been added to Table 1, as follows:

ECG

6 (26.08%)

Low QRS voltages

2 (8.69%)

Epsilon waves

1 (4.35%)

Incomplete right bundle block, Brugada syndrome

1 (4.35%)

Ventricular fibrillation

2 (8.69%)

[Results, Table 1, page 7, line 216]

  1. Paragraph 3.2: please split into separate paragraphs the macroscopic and microscopic findings.

Response. Thank you for this useful comment. According to the reviewer’s comment, the paragraph 3.2. “Anatomo-Pathological Findings of the Heart” has been divided into two separate subsections, as follows:

“3.2.1. Macroscopic Findings

Macroscopically, all the ACM hearts showed global dilatation of the right ventricle (RV) cavity, with transmural muscle loss and thinning of the RV wall (Figure 2), confirmed by transillumination placing the heart wall against a source of light (Figure 3). The heart weight mean ± SD was 410.33 ± 120.67 g, with a range from 92 (in a 5-year-old child) to 664 g. A concomitant cardiac anomaly was observed only in one (4.35%) case that presented a double origin of the right coronary artery from the proper Valsalva sinus.

The RV chamber was generally dilated, presenting bulging and/or aneurysms, accompanied with a thinned RV wall, characterized by fatty scarring pattern (Figure 2). The RV thickness had a mean ± SD of 4.57 ± 2.06, ranging from 1 to 8 mm. The RV free wall was generally markedly thinned and translucent at transillumination (Figure 3). In 9 (39.13%) hearts the RV wall thickness was preserved.

The left ventricle (LV) thickness had a mean ± SD of 17.08 ± 3.25, ranging from 13 to 25 mm. The interventricular septum (IVS) thickness had a mean ± SD of 16.23 ± 3.0, ranging from 9 to 20 mm”.

[Results, page 8, lines 259-272]

3.2.2. Microscopic Findings

Histopathological examination disclosed decreased thickness of the wall of the RV, transmural myocardial cells loss with adipose and fibrous tissue replacement in the RV, in absence of significant coronary artery disease or other known cardiac causes, in all cases (100%). The RV was mostly filled with fatty tissue mixed with fibrous tissue except for the surviving myocardium that was normal or sometimes showed degenerative changes, such as intracytoplasmic vacuoles and myofibrillar loss. The adipose tissue was either disorganized or sometimes had columnar pattern (Figure 4). The fibrous tissue was interstitial fibrosis or replacement fibrosis. Papillary muscles mostly remained normal (78,26% of cases) or were involved by the fatty and fibrous replacement (21.74% of cases).

[Results, page 12, lines 319-328]

  1. In the Discussion, an overview of ACM-associated mutations that have been linked to conduction system impairments could be useful. The authors could speculate and comment about it.

Response: According to the referee’s comments, an overview of ACM-associated mutations that have been linked to conduction system impairments has been added to the text, as follows:

“Mutations in the genes plakophilin-2 (PKP2), desmoplakin (DSP), desmoglein-2 (DSG2), desmocollin-2 (DSC2), junction plakoglobin (JUP), and transmembrane protein 43 (TMEM43) are strongly associated with ACM [10]. The possible ACM-associated mutations linked to conduction system impairments have been little, if at all, investigated. Moreau et al [36] studied an ACM patient with a missense mutation (c.394C>T) in the DSC2 gene, using a zebrafish DSC2 model system. The DSC2 patient-derived pluripotent stem cells were reprogrammed and differentiated into cardiomyocytes, the human-induced pluripotent stem cells cardiomyocytes (hiPSC-CM). Based on the responsiveness to antiarrhythmic drugs, a short QT interval at low heart rate was associated with DSC2 ACM. Recently, Hayashi et al [37] performed Whole-exome sequencing (WES) of 23 probands diagnosed with early-onset (<65 years) of cardiac conduction system diseases analyzing 117 genes linked to arrhythmogenic diseases or cardiomyopathies. Cellular electrophysiological study and in vivo zebrafish cardiac assay showed that two variants in Potassium Voltage-Gated Channel Subfamily H Member 2 (KCNH2) and Sodium Voltage-Gated Channel Alpha Subunit 5 (SCN5A), four variants in Sodium Voltage-Gated Channel Alpha Subunit 10 (SCN10A), and one variant in Myosin Heavy Chain 6 (MYH6) genes resulted from “uncertain significance” to “likely pathogenic” in six probands”.

[Discussion, page 18, lines 470-485]

The following inherent references have been added to the manuscript:

“36.      Moreau, A.; Reisqs, J.; Delanoe-Ayari, H.; Pierre, M.; Janin, A.; Deliniere, A.; Bessière, F.; Meli, A.C.; Charrabi, A.; Lafont, E.; et al. Deciphering DSC2 arrhythmogenic cardiomyopathy electrical instability: From ion channels to ECG and tailored drug therapy. Clin. Transl. Med. 2021, 11, e319, doi:10.1002/ctm2.319.

  1. Hayashi, K.; Teramoto, R.; Nomura, A.; Asano, Y.; Beerens, M.; Kurata, Y.; Kobayashi, I.; Fujino, N.; Furusho, H.; Sakata, K.; et al. Impact of functional studies on exome sequence variant interpretation in early-onset cardiac conduction system diseases. Cardiovasc. Res. 2020, 116, 2116–2130, doi:10.1093/cvr/cvaa010”.

[References, page 24, lines 715-722]

  1. To strengthen the findings and complete the discussion with also evidence in murine models, please revise Cell Junctions in the Specialized Conduction System of the Heart 3109/15419061.2014.905928.

Response: According to the referee’s comments, an overview of evidence of CCS cell junctions in murine models has been added to the text, as follows:

“It is now well known that the CCS cells harbor both mechanical anchoring junctions and electrical gap junction and ion channels. Cell junctions in the specialized cardiomyocytes of the CCS have been studied in in murine models suggesting a potential functional role of anchoring junction components in the CCS. Further studies on the role of junctions specifically within the CCS in the mouse and human will provide insight into uncovering the mechanisms underlying ACM and other human cardiac diseases associated with cell – cell junction defects [48]”.

[Discussion, page 20, lines 558-563]

The following inherent reference has been added to the manuscript:

“48.       Mezzano, V.; Pellman, J.; Sheikh, F. Cell junctions in the specialized conduction system of the heart. Cell Commun. Adhes. 2014, 21, 149–59, doi:10.3109/15419061.2014.905928”.

[References, page 27, lines 833-834]

  1. The adipose and fibrotic replacement in ACM patients’ hearts is not associated with cardiomyocyte transdifferentiation, but the cells responsible for these processes are stromal fibroblastoid cells (doi: 10.1093/eurheartj/ehv579; doi: 10.1161/CIRCRESAHA.115.308136; doi: 10.3390/ijms22052673). In the Discussion, please contextualize your findings taking into account the origin of the fibro-fatty tissue.

Response: The following comments on recent findings on the origin of fibro-fatty tissue in ACM have been added to the text:

“The adipose and fibrotic replacement in ACM patients’ hearts is not associated with cardiomyocyte transdifferentiation, but the cells responsible for these processes are stromal fibroblastoid cells. Lombardi et al [42] reported that in ACM a subset of fibro-adipogenic progenitors (FAPs) expresses desmosome proteins and differentiates to adipocyte upon deletion of the DSP gene. Cardiac mesenchymal stromal cells (C-MSC) are the most abundant cells in the heart, with propensity to differentiate into several cell types, including adipocytes; they contribute to the adipogenic substitution of cardiomyocytes observed in ACM [43]. C-MSC are a source of myofibroblasts and participate in ACM fibrotic remodeling, being highly responsive to ACM-characteristic excess of transforming growth factor-β (TGF-β) [44]”.

[Discussion, page 20, lines 539-547]

The following inherent references have been added to the manuscript:

“42.        Lombardi, R.; Chen, S.N.; Ruggiero, A.; Gurha, P.; Czernuszewicz, G.Z.; Willerson, J.T.; Marian, A.J. Cardiac fibro-adipocyte progenitors express desmosome proteins and preferentially differentiate to adipocytes upon deletion of the desmoplakin gene. Circ. Res. 2016, 119, 41–54, doi:10.1161/CIRCRESAHA.115.308136.

  1. Sommariva, E.; Brambilla, S.; Carbucicchio, C.; Gambini, E.; Meraviglia, V.; Dello Russo, A.; Farina, F.M.; Casella, M.; Catto, V.; Pontone, G.; et al. Cardiac mesenchymal stromal cells are a source of adipocytes in arrhythmogenic cardiomyopathy. Eur. Heart J. 2016, 57, 1835–1846, doi:10.1093/eurheartj/ehv579.
  2. Maione, A.S.; Stadiotti, I.; Pilato, C.A.; Perrucci, G.L.; Saverio, V.; Catto, V.; Vettor, G.; Casella, M.; Guarino, A.; Polvani, G.; et al. Excess tgf-β1 drives cardiac mesenchymal stromal cells to a pro-fibrotic commitment in arrhythmogenic cardiomyopathy. Int. J. Mol. Sci. 2021, 22, 1–16, doi:10.3390/ijms22052673”.

[References, page 27, lines 816-826]

  1. Conclusions: the final message of your work can be better explain. It could be interesting to understand if it could be added to diagnostic criteria or at least to be useful to detect the patients with higher sudden death risks. Are there reliable diagnostic techniques able to visualize the CCS? (For example revise: first in situ 3D visualization of the human cardiac conduction system and its transformation associated with heart contour and inclination 10.1038/s41598-021-88109-7).

Response. Thank you for this useful comment. According to the reviewer’s suggestion, the following sentences have been added to the text:

“Our study of the CCS on serial sections has been carried out post-mortem. Additional clinico-pathological and imaging correlations of the CCS findings would allow to intervene intra-vitam identifying subjects at risk for sudden death. Subsequent actions to avoid the lethal arrhythmic event, such as disqualification from athletic activities and implantation of a portable defibrillator, would be foreseen. Are there reliable diagnostic techniques able to visualize the CCS intra-vitam? Recently, Kawashima et al [53] carried out a specialized physical three-dimensional (3D) computed tomography (CT) micro-dissection serial sections imaging of the CCS in human body. The technique identified that when the cardiac inclination changed from standing to lying, the SAN shifted from the dorso-superior to the right outer position and the AV conduction axis changed from a vertical to a leftward horizontal position. In situ localization of the human CCS provided accurate anatomical localization with morphometric data and useful correlation between heart inclination and CCS rotation axes for predicting variable in the CCS in human living body.

Our histopathological CCS findings will be useful to be paired with advanced future imaging modalities and methodology for further accurate prediction of subjects at risk for sudden death, suffering from ACM or other cardiomyopathies”.

[Discussion, page 23, lines 643-657]

“Future research should focus on application of these knowledge on CCS anomalies to be added to diagnostic criteria or at least to be useful to detect the patients with higher sudden death risks”.

[Conclusions, page 22, lines 669-671]

The following inherent reference has been added to the manuscript:

“53.       Kawashima, T.; Sato, F. First in situ 3D visualization of the human cardiac conduction system and its transformation associated with heart contour and inclination. Sci. Rep. 2021, 11, 1–15, doi:10.1038/s41598-021-88109-7”.

[References, page 27, lines 847-850]

  1. Graphical abstract is missing.

Response:  According to the reviewer’s comments, the Graphical Abstract has been added to the manuscript:

[Graphical Abstract, page 2, lines 41-42]

  1. Please, improve the quality of the images.

Response: According to the reviewer’s comments, the quality of the images has been improved. In particular, Figures 3, 4, 5 and 7 have been replaced.

[Results, Figure 3, page 10, line 288]

[Results, Figure 4, page 11, line 299]

[Results, Figure 5, page 12, line 308]

[Results, Figure 7, page 18, line 436]

  1. Please, check the cited literature and modify it when is not appropriately referred to the text.

Response: All the references have been checked and they are appropriately referred to the text. In response to the comments of both reviewers, the number of references has increased from 42 to 53.

We hope to have exhaustively responded to all comments. Many thanks for the pertinence and accuracy of the comments.

Sincerely,

Giulia Ottaviani, MD, PhD

Reviewer 2 Report

In this manuscript the authors describe mainly histological findings, along with known clinical and genetic data, of patients who died suddenly with a particular focus on patients with arrhythmogenic cardiomyopathy. During anatomical and histological examination, particular focus was put on the conduction system.

While the results are well organised and clearly described, there remains room for improvement in the discussion section. In addition, conclusions on involvement of conduction system and definition of these changes as “congenital”, seem not to be supported by sufficient evidence and we suggest to tone down these conclusions. 

Major comments

  1. Number of the ACM-associated loci.

A recent paper is nicely highlighting the genes that have strong, moderate, or low association with ACM (or ARVC): James et al., Circ Genomics Precision Med 2021. PMID: 33831308. It might be worth citing this manuscript for the current classification of ACM based on genetics

  1. For case #21, which is shown in figure 3, no sections and histological pictures and staining are shown. If these are available with the authors, we would suggest to include in figure 3, in that this would be informative
  2. Discussion, lines 512-519: we would suggest being more careful in stating that the described changes are congenital because they can be detected singe age of 5 years. In fact, to be congenital – it has to be present from birth – by definition – which is in fact not demonstrated by the authors in any way with their work (histological nor mechanistically). Please revise and suggest this as a hypothesis which needs to be further investigated and proven.
  3. The discussion is rather scattered and it mainly describes comparison in numbers and description with other studies. This referees believe the discussion could be better organised e.g. clearly highlighting similarities and differences – along with reasosn for that in a more organised way.

We also suggest not too draw too strict conclusions, rather suggest possibilities.

Please also clearly state limitations of the study (e.g. patient numbers, selection biases, etc…)

Minor

  • Line 71-74 there is a lot of repetition and the text does not read fluent
  • Figure 2 legend. “…towards to…” please correct (no “to”)
  • Figure 3.B reads “…the aright atrium….” Please correct typo (right)
  • Figure 3 C. Pleaase correct the spaces; terms in Latinum should be in italics (e.g. pars membranaces septi”)

Author Response

July 8, 2021

Re: Sudden Unexpected Death Associated with Arrhythmogenic Cardiomyopathy:

Study of the Cardiac Conduction System  (Ms. ID: diagnostics-1271816)

Authors’ Responses to REVIEWERS’ Remarks

We thank the Editor-in-Chief and the two Reviewers for their constructive comments.  Replies to individual comments are stated below each comment. The reviewer’s comments are shown in bold font and our responses are shown in italics.

TO REVIEWER # 2

  1. Number of the ACM-associated loci.

A recent paper is nicely highlighting the genes that have strong, moderate, or low association with ACM (or ARVC): James et al., Circ Genomics Precision Med 2021. PMID: 33831308. It might be worth citing this manuscript for the current classification of ACM based on genetics

Response. Thank you for this useful comment. According to the reviewer’s suggestion, the genes involved in ARVC have been discussed more broadly and the following sentences have been added to the manuscript:

“Recently, an international multidisciplinary ARVC Clinical Genome Resource Gene Curation Expert Panel [10] reappraised all reported ARVC genes. The genes that have strong, moderate, or low association with ACM or ARVC have been highlighted. Of 26 reported ARVC genes, six genes, PKP2, DSP, DSG2, DSC2, JUP, and TMEM43, have strong evidence for ARVC causation. Two genes, DES and PLN, have moderate evidence for ARVC. The remaining 18 genes, such as RYR2, had limited or no evidence for ARVC”.

[Introduction, page 2, lines 60-65]

The following inherent reference has been added to the manuscript:

“[10] James, C.A.; Jongbloed, J.D.H.; Hershberger, R.E.; Morales, A.; Judge, D.P.; Syrris, P.; Pilichou, K.; Domingo, A.M.; Murray, B.; Cadrin-Tourigny, J.; et al. International evidence based reappraisal of genes associated with arrhythmogenic right ventricular cardiomyopathy using the clinical genome resource framework. Circ. Genomic Precis. Med. 2021, 14, e003273, doi:10.1161/CIRCGEN.120.003273”.

[References, page 22, lines 637-640]

  1. For case #21, which is shown in figure 3, no sections and histological pictures and staining are shown. If these are available with the authors, we would suggest to include in figure 3, in that this would be informative

Response. Thank you for this useful comment. The histopathological pictures related to Case # 21, shown macroscopically in Figure 3, are shown in Figure 5. In order to make this concept clearer, the following statements have been added to the inherent legends for figures 3 and 5:

“Figure 3. Gross features of heart’s by transillumination: The opened heart was placed against a source of light. Case # 21, 47-year-old man who died suddenly and unexpectedly; the heart weight was 415 g. A diagnosis of arrhythmogenic cardiomyopathy (ACM) was established at post-mortem examination. The histopathological findings of this case are shown in Figure 5”.

[Results, Legend for Figure 3, page 10, lines 293-296]

“Figure 5. Histopathological sections of the left ventricle (LV) wall. Case # 21, 47-year-old man who died suddenly and unexpectedly. Same case of Figure 3”.

[Results, Legend for Figure 5, page 12, lines 314-315]

  1. Discussion, lines 512-519: we would suggest being more careful in stating that the described changes are congenital because they can be detected singe age of 5 years. In fact, to be congenital – it has to be present from birth – by definition – which is in fact not demonstrated by the authors in any way with their work (histological nor mechanistically). Please revise and suggest this as a hypothesis which needs to be further investigated and proven.

Response · According to the referee’s suggestions, our statements about the possible congenital nature of the fatty-fibrous involvement have been changed, as follows:

 “The fatty-fibrous infiltration frequently detected in ACM represents the morphological substrate for the development of arrhythmic reentry mechanisms, eventually leading to ventricular fibrillation and SUCD. The fatty-fibrous involvement of the CCS has been observed starting from the age of five years in our series, but it is still unknown if it is congenital in nature, i.e., present from birth. The hypothesis that the fatty-fibrous involvement of the CCS is congenital in nature needs to be further investigated and proven. In particular, we suggest that the fatty-fibrous involvement related to ACM was congenital in nature, being detected starting from the age of 5 in our series, and represents the morphological substrate for the development of arrhythmic reentry mechanisms, eventually leading to ventricular fibrillation and SUCD.”

[Discussion, page 19, lines 533-538]

  1. The discussion is rather scattered and it mainly describes comparison in numbers and description with other studies. This referees believe the discussion could be better organised e.g. clearly highlighting similarities and differences – along with reasosn for that in a more organised way.

Response. Thank you for this useful comment. In reply to the suggestions also of the reviewer # 1, the Discussion has been fully edited and organized into the following subsections:

“4.1. Patients’ characteristics

4.2. Genetics

4.3. Anatomo-pathological findings of the heart

4.4. Histopathological findings in the cardiac conduction system”

[Discussion, pages 18-22, lines 445-644]

  1. We also suggest not to draw too strict conclusions, rather suggest possibilities.

Response: In reply also to the comment 8 of the first reviewer, our conclusions now suggest possibilities and future trends, and has been changed as follows:

“The careful examination of the cardiac conduction system on serial sections was crucial in documentingdocumented the adipose and fibrous infiltration of the cardiac conduction system in ACM. Future research should focus on application of these knowledge on CCS anomalies to be added to diagnostic criteria or at least to be useful to detect the patients with higher sudden death risks.”

[Conclusions, page 23, lines 666-669]

Accordingly, the Conclusions in the Abstract have been changed as follows:

“The careful examination of the cardiac conduction system on serial sections was crucial in documenting the fatty-fibrous infiltration of CCS in ACM.Future research should focus on application of these knowledge on CCS anomalies to be added to diagnostic criteria or at least to be useful to detect the patients with higher sudden death risks”.

[Abstract, page 2, lines 34-36]

  1. Please also clearly state limitations of the study (e.g. patient numbers, selection biases, etc…)

Response: The following Limitations subsection has been added to the Discussion:

“4.5. Limitations

Since the hearts were collected mostly from forensic autopsies already performed by the referring centers, it was not possible to obtain a complete clinical record or the genetic tests of the family members. As stated in the Results section, only in a study case the referring center provided the complete genetic testing. Since the Lino Rossi Research Center of the Università degli Studi di Milano, Milan, Italy mostly receive cases of sudden unexpected death, our control group consisted of subjected who died suddenly and unexpectedly for causes different from ACM (Non-ACM group), but might be not representative of the general population. For the same reason, the percentage of SUCD in our series of ACM cases might have been affected by selection bias ”.

[Discussion, page 23, lines 660-667]

Minor

  1. Line 71-74 there is a lot of repetition and the text does not read fluent

Response: The sentence has been reworded for clarity as follows:

“Recently, ARVC has been categorized as arrhythmogenic cardiomyopathy (ACM). which ACM is an arrhythmogenic cardiomyopathy characterized by fibro-fatty replacement of the right ventricle, biventricular of both ventricles or with of the left ventricle involvement which may even exceed the severity of right ventricle involvement.”

[Introduction, page 3, lines 83-86]

  1. Figure 2 legend. “…towards to…” please correct (no “to”)

Response: The sentence has been corrected as follows:

”B) Opened heart by a biventricular section shows severe thinning of RV, which presents a saccular aneurysm of the RV chamber toward to heart’s cardiac apex”

[Results, Figure 2 legend, page 9, line 280]

  1. Figure 3.B reads “…the aright atrium….” Please correct typo (right)

Response: The mistake has been corrected as follows:

“B) The aright atrium wall is so thin to appear devoid of muscle at transillumination”.

[Results, Figure 3 legend, page 10, line 293-294]

  1. Figure 3 C. Pleaase correct the spaces; terms in Latinum should be in italics (e.g. pars membranaces septi”)

Response: The Latin words have been written in italics as follows:

“C) The pars membranacea septi, reference point for the heart dissection to isolate the heart specimen containing the atrio-ventricular junction, has a significantly increased translucency”.

[Results, Figure 3 legend, page 10, line 295-296]

We hope to have exhaustively responded to all comments. Many thanks for the pertinence and accuracy of the comments.

Sincerely,

Giulia Ottaviani, MD, PhD

Round 2

Reviewer 1 Report

The authors gave satisfactory answers to my questions. 

Author Response

Thank you for your favorable comment.